# Long-Term Sodium Deficiency Reduces Sodium Excretion but Impairs Renal Function and Increases Stone Formation in Hyperoxaluric Calcium Oxalate Rats

**DOI:** 10.3390/ijms25073942

**Published:** 2024-04-01

**Authors:** Yuan-Chi Huang, Chan-Jung Liu, Ze-Hong Lu, Ho-Shiang Huang

**Affiliations:** Department of Urology, National Cheng Kung University Hospital, College of Medicine, National Cheng Kung University, Tainan 704302, Taiwan; mrtaiwancj@gmail.com (Y.-C.H.); dragon2043@hotmail.com (C.-J.L.); lthxd@hotmail.com (Z.-H.L.)

**Keywords:** sodium deficiency, sodium transport, aquaporins, hyperoxaluria, kidney stone, calcium oxalate

## Abstract

Excessive sodium intake is associated with nephrolithiasis, but the impact of sodium-deficient (SD) diets is unknown. Hence, we investigated the effects of short- and long-term SD diets on the expression of renal aquaporins and sodium transporters, and thus calcium oxalate (CaOx) crystal formation in hyperoxaluria rats. In a short-term sodium balance study, six male rats received drinking water and six received 0.75% ethylene glycol (EG) to induce hyperoxaluria. After a 30-day period of feeding on normal chow, both groups were treated with a normal-sodium diet for 5 days, followed by a sodium-free diet for the next 5 days. In a long-term SD study (42 days), four groups, induced with EG or not, were treated with normal-sodium water and sodium-free drinking water, alternately. Short-term sodium restriction in EG rats reversed the daily positive sodium balance, but progressively caused a negative cumulative water balance. In the long-term study, the abundant levels of of Na/H exchanger, thiazide-sensitive Na-Cl cotransporter, Na-K-ATPase, and aquaporins-1 from SD + EG rats were markedly reduced, corresponding to a decrease in Uosm, as compared to SD rats. Increased urine calcium, AP(CaOx)index, and renal CaOx deposition were also noted in SD + EG rats. Although the SD treatment reduced sodium excretion, it also increased urinary calcium and impaired renal function, ultimately causing the formation of more CaOx crystals.

## 1. Introduction

Urolithiasis is a common disease with an increasing incidence and worldwide prevalence. Calcium oxalate (CaOx) is the most common component of urolithiasis, especially in idiopathic stone formers, but the pathogenesis of CaOx remains unclear because of its complex contributing factors [1]. Excess dietary salt is widely accepted to be associated with an increasing risk of urolithiasis [2,3,4]. Two possible mechanisms regarding high sodium (Na) intake and adverse changes in urine composition have been proposed: 1. calcium (Ca) excretion is increased by reduced renal tubular reabsorption; 2. urinary citrate is diminished due to loss of bicarbonate [5,6,7]. However, it is interesting that sodium consumption has only been found to be positively correlated with the risk of first-time stone formation in women, and there have been no studies on the role of sodium restriction as an independent factor in reducing the risk of stone formation. Moreover, the impacts of the complete elimination of sodium intake on the risk of CaOx stone formation and renal sodium transporters remain uncertain. These uncertainties leave urolithiasis patients very confused about their optimal sodium intake policy.

The regulation of renal sodium excretion is important for the maintenance of volume homeostasis and urinary concentration. The key sodium transporters responsible for the active transport of sodium chloride (NaCl) and extracellular fluid (ECF) volume homeostasis include the Na-K-2Cl cotransporter (NKCC2), which has also been named BSC1 for bumetanide-sensitive cotransporter, in the thick ascending limb of Henle’s loop [8,9]; thiazide-sensitive Na-Cl cotransporter (TSC), in the distal convoluted tubule [10]; epithelial sodium channel (ENaC) subunits, in the collecting tubules [11]; and type 3 Na/H exchanger (NHE3), in proximal tubules (PT) [12]. Thus, dysregulation of these sodium transporters is believed to participate in lowering CaOx stone risks subsequent to dietary sodium restriction.

The supersaturation of stone promoters, such as calcium and Ox, caused by decreased urine output is a major contributor to the formation of kidney stones. The aquaporins (AQP) are a family of membrane proteins that function as major water channels in mammals, and most of the aquaporin isoforms express in the kidney [12,13,14]. Previous studies have suggested the close link between AQP expression and the formation of Ca-containing kidney stone [5]. Water reabsorption across the PT occurs predominantly by a transcellular pathway which highly depends on AQP-1 in both the apical and the basolateral membranes in this segment. Urabe et al. have found that single nucleotide polymorphisms (SNPs) in AQP-1 were associated with kidney stone formation [15]. Nevertheless, other studies have also noted that the Ca^2+^-sensing receptor (CaSR) activation occurs when there is hypercalciuria, which results in AQP-2 downregulation and polyuria, which inhibits kidney formation [16]. These existing pieces of research evidence have suggested that AQP proteins are closely involved in kidney stone formation. Thus, following dietary sodium restriction, the risk of kidney stone formation may vary further due to variations in AQP expression in the kidney that impact water balance.

Hyperoxaluria induced by 0.75% ethylene glycol (EG) in male rats can induce CaOx crystal accumulation in the kidney without associated metabolic acidosis, and the technique produces a well-accepted animal model which can be used to investigate the mechanisms involved in human kidney stone formation [17,18]. The purposes of the present study are to investigate the short-term and long-term changes in renal function and CaOx stone formation that arise in response to a sodium-deficient (SD) diet, specifically, the complete elimination of sodium intake, in hyperoxaluric rats. We hypothesized that the altered expression of major renal sodium transporters and renal aquaporins that occurs after feeding on an SD diet correlates with the changes in renal sodium metabolism and CaOx accumulation in this animal model.

## 2. Results

### 2.1. Short-Term Sodium and Water Balance Study

First, we investigated the sodium–water balance using a short-term SD diet (5-day). The five days of the experiment included a low sodium diet and tap water free of sodium, in accordance with previous research protocols [19,20]. Figure 1A provides a demonstration of the experimental protocol. In regard to sodium balance (Figure 2A), daily sodium intake was significantly higher and daily sodium balance was significantly more positive in EG rats than in controls within a normal-sodium diet period, whereas daily sodium output showed no significant difference between groups. During the no-sodium diet period, there was no significant change in any of the three shared parameters of the control and the EG rats. With respect to water balance (Figure 2B), increased daily water intake and output were noted in EG rats, but their daily water balance remained unchanged when in the normal-sodium diet period. There was no significant difference in any of the three parameters shared between these two groups during the no-sodium treatment.

During the normal-sodium 5-day period (Figure 3A), more positive cumulative sodium balance was found from days 32 to 35 in EG rats than in the controls, whereas these levels were similar during the no-sodium diet period. More negative cumulative water balance was found in EG than control on days 41 and 42 during no-sodium diet period, whereas it was similar between control and EG rats when in the normal-sodium diet period (Figure 3B). There were no significant differences in renal function, as demonstrated by proteinuria and creatinine clearance (CCr) (Appendix A).

### 2.2. Effect of Long-Term SD Diet on Renal Function

Next, we used a long-term (42-day) SD diet to examine the effects of chronic sodium depletion on renal function and kidney stone formation; the experimental protocol is depicted in Figure 1B. A long-term SD diet should be administered for a minimum of three weeks, according to the prior research [21]. Subsequently, 42 days has been found to be the best length of time, as described in our previous study [22], particularly when it comes to successful stone production in studies utilizing 0.75% EG in distilled water to create hyperoxaluric kidney stone rats. Together, we ultimately decided on a 42-day SD diet as the long-term protocol. Daily water intake increased significantly in EG and SD + EG groups when compared with the controls and SD group. Comparing with the control and EG groups, rats in all SD-treated groups treated for 42 days evinced a decreased urinary sodium excretion rate (UNaUv) and fractional excretion of sodium (FENa) (Table 1). Plasma sodium levels were unchanged in all experimental groups. Although this finding may cast doubt on the efficacy of the SD diet, previous studies investigating its effects have also found that plasma sodium remained unchanged despite compensatory changes in urine sodium [23,24]. After EG induction, total 24 h urine output significantly increased in EG and SD + EG rats. Accompanied by this increased urine output, plasma osmolality was also increased, but urine osmolality decreased significantly, which is suggestive of renal function deterioration. Serum creatinine and urea levels increased, but CCr decreased in EG and SD + EG groups, when compared with control and SD groups, respectively, and to a significant degree.

### 2.3. Effect of Long-Term SD Diet on CaOx Stone Formation

There were no significant differences between groups as to final body weight and the changes in body weight during experiments (Table 2). In all EG-treated rats, increased urinary oxalate, urinary tubular enzymes, and lipid peroxides were noted. CaOx crystal accumulation in the kidney also increased in EG (grade II–III) and SD + EG (grade III) groups (Table 2, Figure 4). Consistent with the changes in kidney stone formation, kidney weight, AP(CaOx) index, and urinary Ca/Cr ratio increased significantly in all EG-treated groups.

We further reviewed the detailed renal histology for the EG-treated groups, as described in Appendix A. For the EG group, the crystals diffusely deposited in renal tubules, distributing throughout the cortex and medulla regions. Significant dilated tubules on the corticomedullary junction were present in the SD + EG group, and diffused tubules showed massive accumulations of calcified crystals. Next, we measured the degree of apoptosis in the four groups. The number of TUNEL-positive cells increased dramatically in the SD + EG group, regardless of whether it was compared to the EG group or the SD group (Figure 5).

### 2.4. Changes in the Abundance of Major Renal Sodium Transporters

The expression levels of major renal sodium transporters were examined in EG and SD + EG rats and compared with those in age- and time-matched control rats (Figure 6). Immunoblotting revealed that the abundance of NHE3 was significantly decreased in the SD + EG rats when compared with the controls and EG rats, whereas NHE3 abundance increased significantly in SD group. Similar changes can also be found in the α-subunit of Na-K-ATPase, TSC, and ATPase. The original blots are presented in Appendix A.

### 2.5. Changes in AQP Abundance

Finally, we examined the abundance levels of AQP1, AQP2, and AQP3 through immunoblot analysis, although the beta-actin in the Western blot of AQP was uneven. The AQP family result should be more appropriately placed in the Appendix A, even though we have included the quantification of the immunoblot analysis in comparison to the control (Appendix A). The abundance levels of AQP1, AQP2, and AQP3 were markedly reduced in EG rats, and there were no significant changes in SD rats when compared with the controls. With respect to SD + EG rats, their AQP3 abundance decreased when compared with EG rats. The original blots are presented in Appendix A.

## 3. Discussion

The present study demonstrated that a normal-sodium diet will cause daily sodium balance to become more positive in EG rats, and that EG rats increased their water intake to compensate and maintain a daily water balance in the normal range during the normal-sodium diet period. However, when in the SD diet period, impaired renal water handling caused water imbalance in the last two-day SD treatment in the EG rats. In a long-term treatment study, FENa and urinary sodium excretion decreased in SD rats, and abundance levels of renal NHE3, TSC, and α-EnaC increased significantly. These adaptive regulations of sodium transporters, a response to a sodium-deficient diet in SD rats, are compatible with the findings of a previously published study [25]. However, SD + EG rats had tubular damage which was more profound, and this may have resulted in decreased Ccr and abundant renal NHE3, TSC, ATPase, and AQP-1. Meanwhile, profound tubular damage incidence in SD + EG rats also led to increased FENa, with a 1.8-fold increase, and UNaUv, which increased by 69.7%, when compared with SD rats, whereas FENa only increased by 33.3%, and UNaUv increased by 11.7% in EG rats, when compared with control rats. Therefore, SD + EG rats had a more serious sodium loss in all experimental groups.

It is widely acknowledged that NHE3 is the most important sodium transporter responsible for sodium reabsorption within the kidney, among all sodium transporters and cotransporters, whereas NHE3 in the PT is directly and indirectly responsible for more than 50% of glomerular filtered sodium reabsorption [26]. The activation of NHE3 facilitates transcellular sodium reabsorption, in conjunction with the Na-K-ATPase in the basolateral plasma membrane. Meanwhile, AQP-1, -2, and -3 abundance levels are tightly regulated in response to renal deterioration and volume homeostasis [14,27]. The activation of the AQP family results in significant polyuria and impaired urinary-concentration capacity. Firstly, we investigated the physiological changes in the kidneys during the formation of stones, as shown in the groups treated with EG. The EG group exhibited statistically significant increases in urine output, plasma osmolality (Table 1), water intake, FE_Na_, and FE_K_ (Appendix A), as well as significant decreases in urine osmolality and the urine/plasma osmolality ratio, in comparison to the control group. The SD + EG group had significant increases in water intake, urine output, plasma osmolality, and FE_Na_, and significant decreases in urine osmolality and the urine/plasma osmolality ratio, in comparison to the SD group. The sodium balance study revealed that the EG group had significantly higher sodium accumulation than the control group (Figure 2). All of these physiological changes suggest that stone formation enhances diuresis and natriuresis, both of which cause volume depletion. While AQP-1, -2, and -3 were all downregulated in the EG group, when compared to the control on immunoblots, the other sodium transporters did not show any significant changes. However, when comparing the SD group and the SD + EG group, α-ENaC, NHE3, TSC, and ATPase were all downregulated. Previous studies have used NHE3 knockout mice to investigate the physiological function of NHE3 in the kidney [28,29,30,31], and these mice increased their fluid intake, urine output, and urinary Na^+^ and K^+^ excretion, along with demonstrating decreased urine osmolality, determinations which are consistent with our findings in the EG-treated groups. The downregulation of NHE3 was also possibly connected to the concurrent changes in α-ENaC and ATPase observed in our results, although more research is needed to confirm this tentative theory. Nevertheless, it is still unclear why the AQP family was all downregulated in the EG group. Under physiological conditions, a dehydrated subject will experience increased sensations of thirst and plasma osmolality, allowing the kidney to excrete hypertonic urine. This reduction in the net loss of water will allow the plasma osmolality to return to the normal range. However, the EG-treated groups had increased plasma osmolality and hypotonic urine formation, representing an imbalanced physiological condition. First, we used T_c_H_2_O (solute-free water reabsorption) to assess the clearance of free water; a positive free-water clearance is associated with the diuresis action, and the results of T_c_H_2_O were all positive in EG-treated groups. Meanwhile, a water-balance test showed no significant difference in cumulative water between the EG and control groups. All of these pieces of evidence suggested that the EG-treated kidney acts to compensate for diuresis by downregulating the AQP family, though further studies are still required to support this theory. 

Next, we investigated the physiological changes after dietary sodium restriction, as shown by the groups treated with SD. Compared with the control, the SD group had a significant increase in urine osmolality and in the ratio of urine/plasma osmolality, and significant decreases in FENa and U_Na_ × Uv were also found. The SD + EG group had a significant increase in plasma urea and T_c_H_2_O but decreases in FENa and Una × Uv, compared to the EG group. With Western blot analysis, we also noted significant increases in α-ENaC, NHE3, and TSC within the SD group, in contrast to the control group. All of these changes point to the kidney’s inhibition of natriuresis as a means of compensating for low dietary sodium intake. Additionally, we discovered that the SD + EG group’s renal functions had significantly declined in comparison to those of the EG group, as evidenced by elevated plasma urea, urine β-galactosidase (GAL), and N-acetyl-β-glucosaminidase (NAG) (Table 2). These findings suggest that SD diet could put greater pressure on renal function in subjects with kidney stone disease.

Based on these interpretations of our findings regarding the physiological changes in response to EG and SD, respectively, the implications of our study can advance our knowledge of how stones form in hyperoxaluric rats. We observed a significant decrease in the expressions of all AQP transporters but no significant changes in sodium transporters after EG treatment when compared to the controls. This adaptation of the kidney in the EG group caused diuresis and exacerbated renal function. In SD conditions, EG treatment not only downregulated AQP-1, -2, and -3, but also inhibited sodium transporters, including NHE3, TSC, and ATPase. These molecular changes in the SD + EG group led to severe diuresis and sodium loss, resulting in the worst renal function recorded among all experimental groups.

It is well-accepted that high sodium intake leads to increased urine calcium excretion and decreased urine citrate, both of which are associated with increased urolithiasis risk [32]. Our previous study also revealed that a high-sodium diet induced massive CaOx crystal accumulation [33]. Dietary sodium restriction is the recommended dietary intervention for stone formers. However, the novel finding of the present study is that excessive sodium restriction didn’t diminish CaOx crystals. Instead, it exacerbated CaOx crystal accumulation. In addition to increased kidney crystal deposits in the SD + EG group, urine Ca/Cr and GAL were actually significantly higher than the EG group, which indicated that excessive sodium restriction cannot provide a beneficial effect in EG rats. We propose the possible explanations for this novel finding as follows. First, the SD + EG rats may have profound renal tubule damage via the increasing OS mentioned above. Profound renal tubular damage will enhance CaOx crystal deposition in the renal tubule [34,35]. Second, most calcium and sodium were reabsorbed in the PT, and the reabsorption of calcium in the PT depends on two mechanisms: (1) accompanied by sodium and fluid absorption, the concentration of calcium in the tubular fluid increases, thereby enhancing the driving force for its passive absorption; (2) positive voltage in the lumen serves as an additional driving force for passive calcium absorption. In each instance, the calcium absorption parallels that of sodium. Therefore, the decreased sodium absorption in SD + EG rats prompted by decreased abundance of AQP1, NHE3, and Na-K-ATPase in PT may result in decreased calcium reabsorption. This can also explain the results of the present study showing high AP(CaOx) index and hypercalciuria in the SD + EG group, as compared to the EG group. Further studies are warranted to confirm our hypothesis. Expert advice from nephrologists and clinical biochemical colleagues is required in this regard.

We recognize several limitations of the current study. The scope of our analysis was restricted by the inability to verify our hypothesis with human data. The absence of human results limited our findings given the concern over whether the findings could be extended to human cases. The fact that human data were not included restricted our conclusions to concerns pertaining to rodents alone. In addition, reduced dietary sodium intake is a more appropriate clinical scenario for the study of impacts on kidney stone disease, because severe dietary sodium depletion is uncommon in real life. Finally, we have noticed that the beta-actin expression in our results was higher in the EG-treated group, even under the same loading conditions, and this result may be possibly related to the fact that beta-actin has been identified as a CaOx crystal-binding protein on renal tubular epithelial cells [36]. With the same quantity of protein, the appearance of beta-actin on immunoblots in the group with kidney stones would be still more obvious. In order to provide a more accurate quantification of the immunoblot analysis, we quantified the intensities of all bands densitometrically, relative to the control.

Conclusively, the present study demonstrated that excessive sodium restriction in hyperoxaluria rats resulted in increased urinary CaOx supersaturation, as well as increased OS. Our findings indicated that most major kidney sodium transporters increased in the SD group, which significantly attenuated urinary sodium excretion. On the contrary, the SD + EG group showed decreased expression of AQP transporters and major sodium transporters, such as NHE3, TSC, and ATPase, compared to the EG group. This led to increased urinary sodium loss and diuresis, and all of these physiological alternations are possibly related to the deterioration of renal function and retention associated with CaOx kidney stones. These novel insights as to the influence of sodium restriction on CaOx formation and renal function may provide a different strategy of dietary intervention for kidney stone formers.

## 4. Materials and Methods

### Animals

Thirty-six male Wistar rats, 6 weeks old, and weighing between 200 and 250 g, were obtained from the colonies of BioLASCo Taiwan Co., Ltd. (Taipei, Taiwan). The rats were housed 4 per cage in an environmental chamber maintained at 24 °C, and with 50% relative humidity, with free access to water and food. Lighting was activated automatically from 08:00 a.m. to 08:00 p.m. Both standard laboratory chow and water were provided each day. All rats were sacrificed with isoflurane (Attane 074416, Panion & BF Biotech, Taipei, Taiwan). All of the environmental protocols were approved by the Institutional Animal Care and Use Committee of National Taiwan University. The study was conducted according to ARRIVE guidelines.

Urine and Plasma Chemistries

Urine Cr and protein were measured on an Arkray RT-4010 urine autoanalyzer (Arkray, Kyoto, Japan). Urine and plasma urea, sodium, potassium, magnesium, and calcium were measured using a Spotchem EL SE-1520 (Arkray, Kyoto, Japan). Urine and plasma osmolality were measure using an osmometer (Micro-Digital Osmometer 7iM, Loser, Berlin, Germany). Urine citrate and Ox were both measured by colorimetric assay kit (Citric Acid and oxalic acid, Elabscience, Wuhan, China).

Short-term Sodium Balance Study

In the sodium balance study, each rat was placed in an individual metabolic cage and provided with a standard rat diet (PMI Nutrition International, St. Louis, MO, USA, contain 0.4% sodium) and 0.75% EG in their drinking water, provided ad libitum. The short-term sodium balance protocol was a modified version of that described in DiBona GF and Sawin LL [19], and consisted of two groups: 1. the control group (received drinking water, *n* = 6); 2. EG-induced hyperoxaluric CaOx kidney stone group (*n* = 6). The animals were treated for 30 days in each group with normal rat chow (PMI Nutrition International, Purina Mills, Inc., Gray Summit, MO, USA), and then entered the following experimental protocols. The scheme of each group consisted of the provision of 2 different levels of dietary sodium intake, each for 5 days and in the following order: normal sodium (days 31 to 35, received SD diet (ICN Biomedicals, Aurora, OH, USA) and 50 meq L^−1^ NaCl drinking solution) and no-sodium (days 38 to 42, received SD diet and tap water, 0 meq L^−1^ NaCl). At the end of a dietary period (day 35), the next diet in the sequence was instituted, and balance measurements started 2 days later, after the immediate transient changes and during more steady-state conditions (day 38).

Sodium concentrations in drinking fluid and urine were determined with flame photometry (Eppendorf FCM 6341, Geratebau, Germany). Calculations were made as follows: Daily sodium intake = (milliliters drinking solution consumed per day) × (sodium concentration of drinking solution); Daily sodium output = (urine volume per day) × (urine sodium concentration); Daily sodium balance = daily sodium intake − daily sodium output. Cumulative sodium balance per diet period was calculated as continuous summation of daily sodium balance within each 5-day dietary period. Similar calculations were made for water balance.

Effect of long-term SD diet feeding on CaOx crystal deposition

Male Wistar rats were housed under constant temperature and light/dark cycle (light from 07:00 to 18:00) conditions. The rats were randomized into four groups, matched for age, as described in the following (*n* = 6 in each group): (1) the control group, (2) the EG group, (3) the SD group, and (4) the SD + EG group. All four groups of rats were fed with a sodium-deficient diet (0.006% sodium, ICN Biomedicals, Aurora, OH, USA). The control group received 50 mEq/L NaCl in drinking water, whereas the SD group received sodium-free water (no NaCl). In EG and SD + EG groups, the rats drank 50 mEq/L NaCl water and sodium-free water with 0.75% EG, respectively. The animals were individually kept in metabolic cages 3 days before sacrifice. At the conclusion of the 6-week diet challenge, the rats were euthanized.

Daily urinary output and water intake were determined throughout the study. Twenty-four-hour urine samples were collected 1 day before sacrifice with the presence of thymol and 0.5 mL 12 N hydrogen chloride (HCL) in the container, and urinary volume was recorded. Osmolality, creatinine, and sodium concentrations in urine and serum specimens were measured at the central laboratory. The urine samples were also used to determine the oxalate (Sigma, St. Louis, MO, USA), citrate (R-Biopharm GmbH, Darmstadt, Germany), calcium, and magnesium levels (measured at the central laboratory). Urinary NAG and GAL were determined, using a method previously described [37]. Urinary lipid peroxide was examined by measuring thiobarbituric acid-reactive substances (TBARS, ZeptoMetrix, Buffalo, NY, USA), and malondialdehyde (MDA, OxisResearch, Portland, OR, USA). All urine samples were assayed in duplicate.

Urinary supersaturation with respect to CaOx was assessed by using the index proposed by Tiselius et al. [38], in which the ion activity product is estimated by following formula: 4067 × Calcium^0.93^ × Oxalate^0.96^ × (Citrate + 0.015)^−0.60^ × magnesium^−0.55^ × Volume^−0.99^. Urinary calcium, oxalate, citrate, and magnesium are expressed in millimoles per 24 h, and urine volume (V) in liters, for the purposes of the calculation. The presence of CaOx crystals in the kidneys was examined and graded, as previously described [22]. In brief, tissue slices were prepared for staining with haematoxylin and eosin (H.E.) to check the pathological changes, and then examined by polarizing microscopy (Olympus BH-2 model, Tokyo, Japan) to check for the presence of calcium oxalate crystals in the kidneys. Each kidney was scored semi-quantitatively into 4 grades (0, I, II, or III), ranging from ‘no crystal deposit (0)’ to ‘massive crystal deposits (III)’. This scoring system has been used extensively in the previously published studies to quantitatively assess the degree of kidney stone formation [22,37,39].

Western blot analysis

After homogenization, kidney specimens were added to a mixture of RIPA protease inhibitors. Using a protein assay kit (Bio-Rad Laboratories, Hercules, CA, USA), the protein concentration of the supernatant was determined following centrifugation. The samples underwent a series of steps, including mixing with SDS loading buffer, boiling, electrophoresing in 10% SDS-PAGE gels, and membrane transfer to PVDF. Membranes (Amersham, Buckingham, England, UK) were incubated with primary antibodies after being blocked with blocking buffer overnight at 4 °C. Primary antibodies for NHE3, TSC, and BSC-1, as well as α, β, and γ-ENaC, were purchased from Alpha Diagnostic International (San Antonio, TX, USA); beta-actin was purchased from Sigma Chemical Co. (St. Louis, MO, USA), and AQP was purchased from Alomone Labs Ltd. (Jerusalem, Israel). After washing, the membrane was then incubated with horseradish peroxidase-conjugated secondary antibodies (1:200; Vector, Burlingame, CA, USA) for 1 h at room temperature. Immunoreactive protein detection was performed with an enhanced chemiluminescence detection system (PerkinElmer, Waltham, MA, USA). The immune complex was visualized by using a commercial ECL kit (Amersham) after development of the film (Kodak). The density of the bands was semi-quantitatively determined by densitometer with an image analytic system (Alpha Innotech, San Leandro, CA, USA). The level of each protein was assessed relative to the amount of β-actin.

Statistical analyses

Numerical data are presented as the mean ± standard error of the mean. Differences between subgroups were analyzed using an unpaired *t*-test or one-way ANOVA, and Duncan’s multiple-range test was used to compare subgroups. The Pearson product–moment correlation method was used to determine the correlation coefficient (γ value) in the acute sodium oxalate infusion experiment. Differences were regarded as significant at *p* < 0.05.

Ethics approval

All animal experiments were conducted under protocols approved by the Institutional Animal Care and Use Committee of National Taiwan University and were performed in compliance with the Guide for the Care and Use of Laboratory Animals (Published by National Academy Press, Washington, DC, USA, 1996), with due consideration to the minimization of pain and suffering.

## Figures and Tables

**Figure 1 ijms-25-03942-f001:**
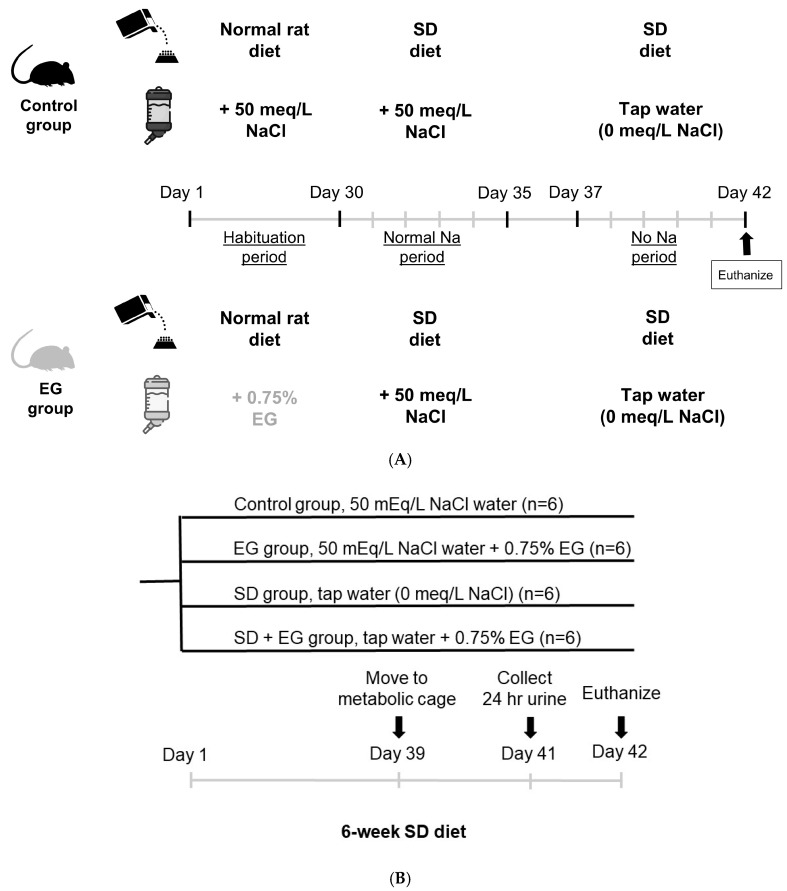
(**A**) Protocol of the short-term sodium balance study. The control group (received drinking water) and the EG-induced hyperoxaluric CaOx kidney stone group (received 0.75% EG in the drinking water) were fed with normal rat chow for 30 days. Then, a normal-sodium diet (Day 31~35) and a subsequent no-sodium diet (Day 38~42) were given, each for 5 days. (**B**) Protocol of the study of chronic sodium depletion on renal function and kidney stone formation: The rats were randomized into four groups: (1) the control group, (2) the EG group, (3) the SD group, and (4) the SD + EG group. All of the rats received an SD diet. SD, sodium-deficient diet; EG, ethylene glycol; Na, sodium; NaCl, sodium chloride.

**Figure 2 ijms-25-03942-f002:**
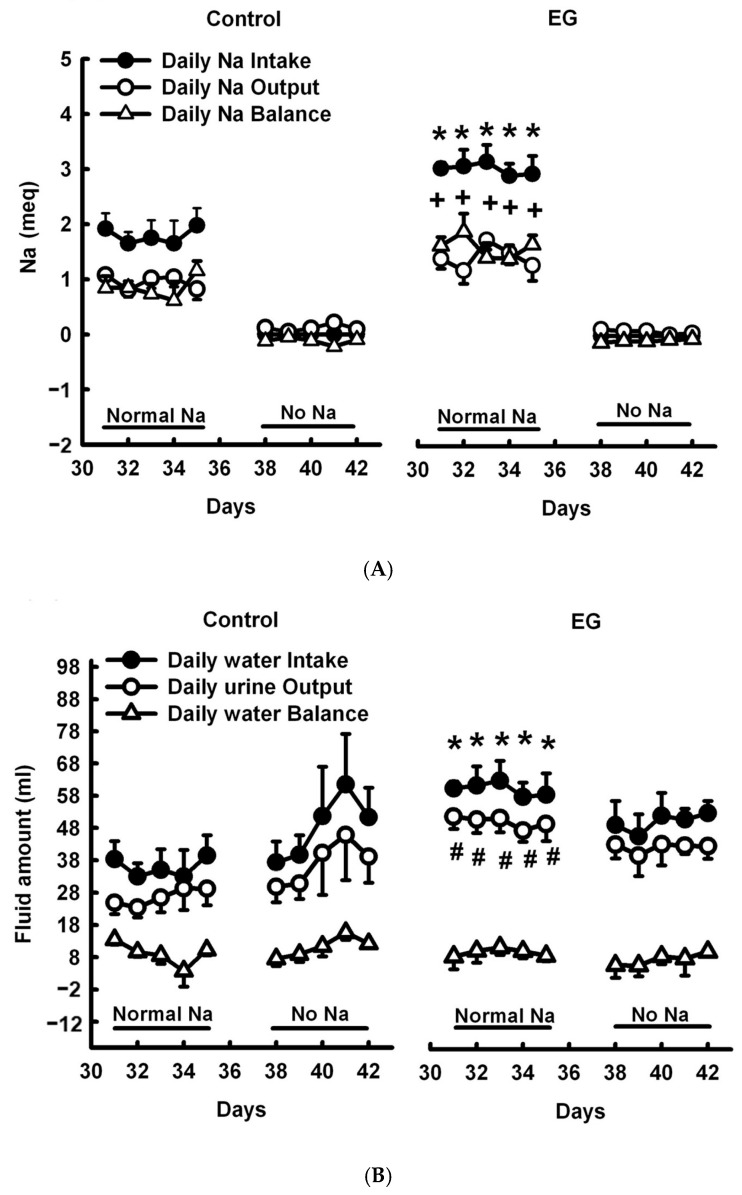
(**A**) Daily sodium intake, daily sodium output, and daily sodium balance during normal-sodium and no-sodium intake periods in the control (left panel) and ethylene glycol (EG)-treated rats (right panel). (**B**) Daily water intake, output, and balance during normal-sodium and no-sodium intake periods in the control and EG-treated rats. EG, ethylene glycol; Na, sodium. * *p* < 0.05 when comparing daily intake between the two groups. # *p* < 0.05 when comparing daily output between the two groups. + *p* < 0.05 when comparing daily balance between the two groups.

**Figure 3 ijms-25-03942-f003:**
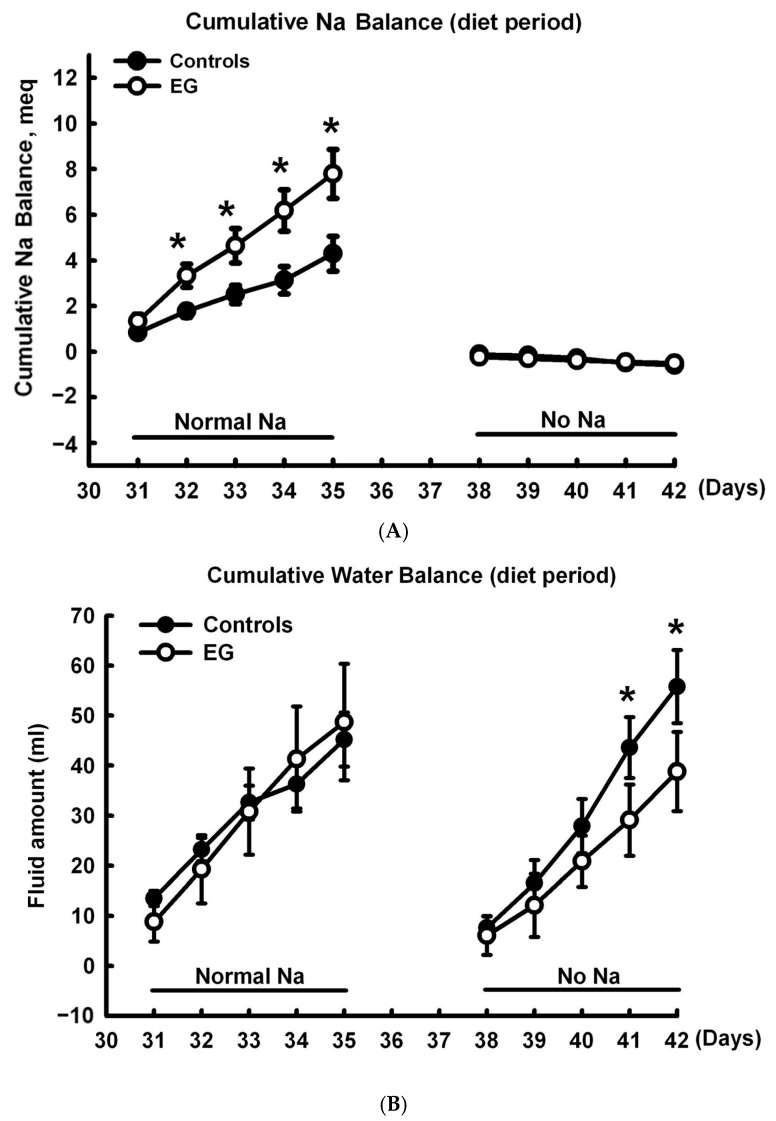
(**A**) Cumulative sodium balance and (**B**) cumulative water balance per diet period during normal diet and sodium-deficient diet (SD) periods in the control and EG-treated rats. EG, ethylene glycol; Na, sodium. * *p* < 0.05.

**Figure 4 ijms-25-03942-f004:**
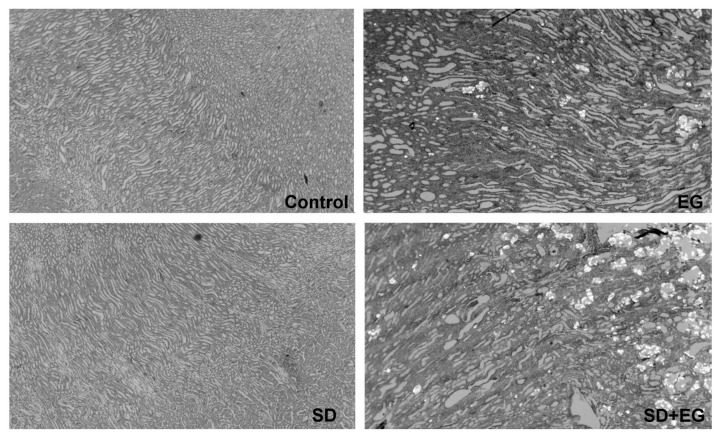
Representative micrographs demonstrate the calcium oxalate (CaOx) calculi deposit in a kidney from each group. Reduced from ×40. SD, sodium-deficient diet; EG, ethylene glycol.

**Figure 5 ijms-25-03942-f005:**
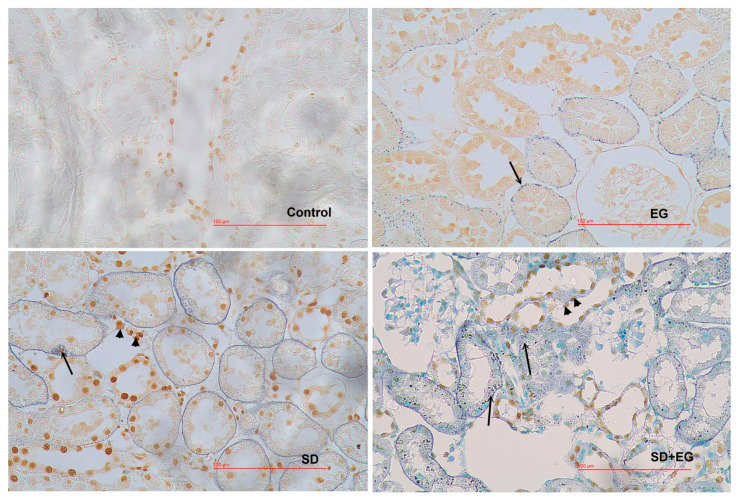
Apoptotic cells in the kidneys. Apoptotic cells (indicated by black arrow) were evaluated using TUNEL staining after a nitro blue tetrazolium (NBT) perfusion test had been performed to localize de novo superoxide production (an insoluble blue formazan derivative, indicated by black arrowhead) in the kidney of all four groups. Reduced from ×400 magnification. The relationship between changes in scores of formazan particles and TUNEL staining positive cells in kidney tissues shows a positive correlation, and *p* < 0.05. SD, sodium-deficient diet; EG, ethylene glycol.

**Figure 6 ijms-25-03942-f006:**
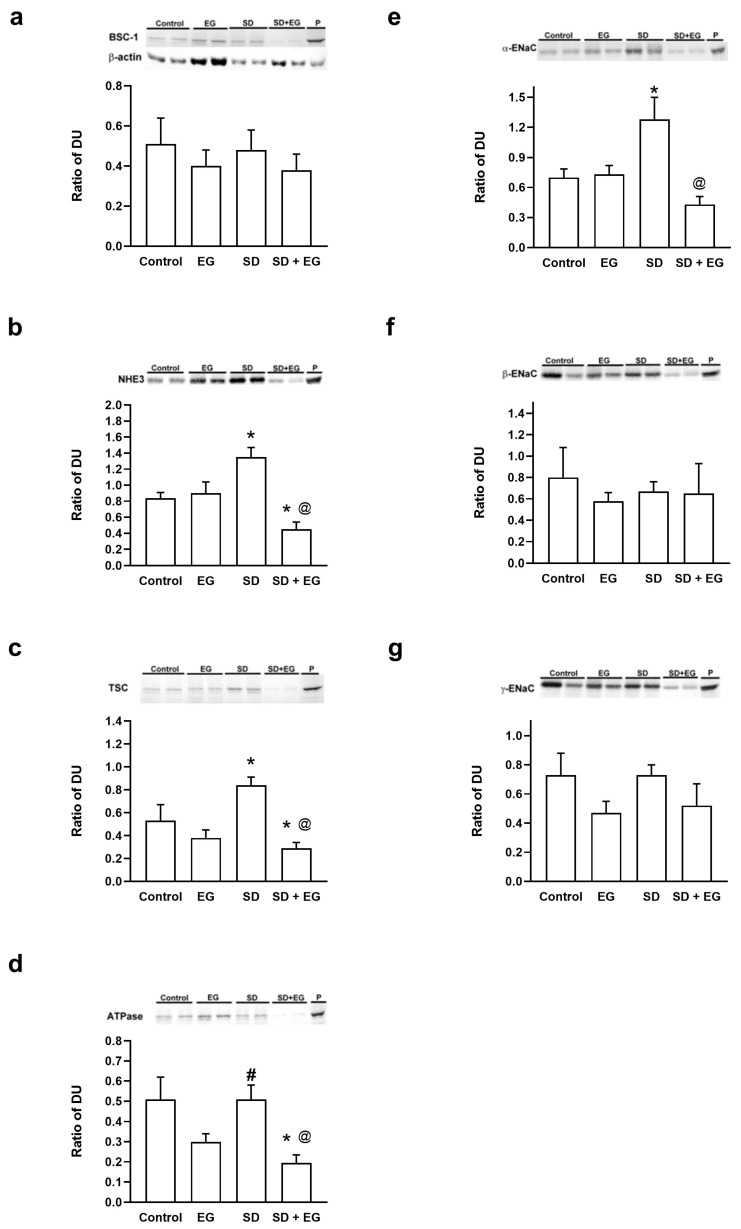
Representative semiquantitative immunoblotting of kidney: (**a**) BSC-1, (**b**) Type 3 Na^+^/H^+^ exchanger (NHE3), (**c**) TSC, (**d**) Na^+^-K^+^-ATPase, (**e**) α-ENaC, (**f**) β-ENaC, and (**g**) γ-ENaC. DU, densitometric unit; P, positive control; BSC-1, bumetanide-sensitive cotransporter type 1; TSC, thiazide-sensitive Na-Cl cotransporter; ENaC, epithelial sodium channel. * *p* < 0.05, compared to the control group. ^@^
*p* < 0.05, compared to the SD group. ^#^ *p* < 0.05, compared to the EG group.

**Table 1 ijms-25-03942-t001:** Renal functions data (42-day study group).

	Groups	Control	EG	SD	SD + EG
Parameters					
Water intake, mL/day		25.6 ± 4.6	31.8 ± 3.3	29.6 ± 2.4	45.0 ± 4.9 *
Urine amount, mL/day		19.9 ± 1.5	27.1 ± 1.8 *	16.2 ± 1.8	39.7 ± 1.4 *^@^
Urine protein, mg/day		24.1 ± 4.2	55.1 ± 9.5 *	21.3 ± 4.0	66.4 ± 9.4 *^@^
C_Cr_, mL/min/kg		2.7 ± 0.9	1.7 ± 0.3	3.8 ± 0.7	1.4 ± 0.4 ^@^
Posm, mosmol/kg/H_2_O		297.4 ± 4.6	337.0 ± 5.5 *	292.3 ± 4.7	333.2 ± 6.0 *^@^
P_Na_, mmol/L		150.3 ± 2.6	156.5 ± 1.6	147.4 ± 3.9	159.3 ± 1.7
P_Cr_, mg/dL		0.31 ± 0.01	0.57 ± 0.05 *	0.31 ± 0.01	0.82 ± 0.15 *
P_Urea_, mg/dL		16.7 ± 2.9	40.0 ± 2.2 *	19.4 ± 1.5	54.3 ± 6.6 *^#^
Uosm, mosmol/kg/H_2_O		1026.8 ± 188.0	626.8 ± 20.4 *^@^	1380.9 ± 132.5 *^#^	702.8 ± 29.1 *^@^
U/Posm ratio		2.9 ± 0.1	1.9 ± 0.0 *	4.7 ± 0.4 *	2.1 ± 0.1 *^@^
FE_Na_ (%)		0.5 ± 0.1	0.7 ± 0.1	0.1 ± 0.0 *^#^	0.3 ± 0.1 *^#@^
U_Na_ × U_v_, μmol/min/kg		1.0 ± 0.1	1.1 ± 0.2	0.3 ± 0.1 *^#^	0.5 ± 0.0 *^#@^
T_c_H_2_O, μL/min/kg		78.6 ± 10.4	52.1 ± 3.9 *	108.3 ± 19.4 ^#^	77.6 ± 8.8 ^#^

SD, sodium-deficient diet; EG, ethylene glycol; P, plasma; Posm, plasma osmolality; P_Cr_, plasma creatinine; P_Urea_, plasma urea level; Uosm, urine osmolality; FE_Na_, fractional excretion of Na; U_Na_ × U_v_, rate of urinary Na excretion; T_c_H_2_O, solute-free water reabsorption. * *p* < 0.05, compared to control group. ^@^ *p* < 0.05, compared to SD group. ^#^ *p* < 0.05, compared to EG group.

**Table 2 ijms-25-03942-t002:** Results of body weight, left-kidney weight, creatinine clearance, urinary oxalate levels, urinary lipid peroxides, and tubular enzymuria.

	Groups	Control	EG	SD	SD + EG
Parameters					
BW, gm		380.4 ± 4.2	397.2 ± 4.4	393.1 ± 2.5	396.7 ± 3.6
ΔBW, gm		+5.1 ± 1.0	+7.2 ± 1.7	+8.0 ± 2.8	+5.0 ± 0.5
Food intake, g/day		26.3 ± 3.5	18.5 ± 2.9	27.7 ± 3.4	16.2 ± 2.6 *
Kidney weight, gm		1.8 ± 0.1	2.6 ± 0.1 *	1.8 ± 0.1	2.7 ± 0.1 *
AP (CaOx)index		4.8 ± 0.9	21.2 ± 6.1 *	4.5 ± 1.0	24.9 ± 7.3 *
Urine Ca/Cr		0.3 ± 0.1	0.2 ± 0.0	0.1 ± 0.0 *	0.8 ± 0.2 *^#@^
Urine oxalate, mg		6.8 ± 2.2	68.3 ± 12.2 *	4.4 ± 1.6	75.6 ± 23.3 *
GAL, mU/mg		3.4 ± 0.9	8.5 ± 2.1 *	2.3 ± 0.2	38.9 ± 16.1 *^#^
NAG, mU/mg		4.0 ± 0.6	20.3 ± 5.8 *	3.3 ± 0.2	43.9 ± 12.8 *^#^
Urine MDA, μmole/g		39.0 ± 8.3	402.8 ± 157.9 *	53.0 ± 4.9	514.0 ± 208.5 *
Urine TBARS, μmole/g		156.6 ± 28.5	632.9 ± 229.4 *	199.5 ± 16.7	883.9 ± 277.6 *
Renal calculi (score)		0	II–III	0	III

SD, sodium-deficient diet; EG, ethylene glycol; BW, body weight; Cr, creatinine; ΔBW, BW change between basal period and the end of experiment; GAL, β-galactosidase; NAG, N-acetyl-β-glucosaminidase; MDA, malondialdehyde; TBARS, thiobarbituric acid reactive substances; AP (CaOx) index, the ion activity product of CaOx. * *p* < 0.05, compared to control group. ^@^ *p* < 0.05, compared to SD group. ^#^ *p* < 0.05, compared to EG group.

## Data Availability

The data presented in this study are available on request from the corresponding author due to privacy.

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
