# Peer review of "Long-Term Sodium Deficiency Reduces Sodium Excretion but Impairs Renal Function and Increases Stone Formation in Hyperoxaluric Calcium Oxalate Rats"

_ijms, 2024, doi:10.3390/ijms25073942_

Round 1
Reviewer 1 Report
Comments and Suggestions for Authors
The manuscript offers a comprehensive exploration of the effects of short-term and long-term sodium deficiency on kidney function, calcium oxalate stone formation, and sodium transporters in hyperoxaluric rats. The study is well-organized, with meticulous detailing of methods and results, establishing a robust research foundation. The paper is intriguing and innovative, with only minor areas for improvement.
1) While plasma Na levels remained unchanged in all experimental groups, clarification on how sodium deficiency was confirmed is needed. Please provide an explanation.
2) For clarity in understanding the study protocols in the Results section, it is recommended to briefly describe them in the legend of Fig. 1 and define all abbreviations used.
3) In the discussion section, enhance clarity on the implications of the study findings. Specifically, elaborate on how the observed changes in sodium transporters and kidney function contribute to our understanding of stone formation in hyperoxaluric rats.
4) Include a section on the study limitations.
5) Ensure consistency in terminology and abbreviations throughout the manuscript.
Comments on the Quality of English LanguageCareful proofreading is recommended to address minor grammatical errors and improve overall language quality.
Reviewer 2 Report
Comments and Suggestions for Authors
The topic of the article creates the premises for a very interesting discussion. As we all know the excessive sodium intake correlates with an increased risk of nephrolithiasis. The authors want to evaluate what are the risks on nephrolithiasis in sodium-deficient diets. The design of the study is well done and serve the purpose. However, despite the design is sound, there are some flaws that require to be corrected. Some of them are problems related to form and other are problems related to content.
In the Discussion section, line 297: "This net loss of water will allow the plasma osmolality to return to the normal range. " or the reduction of net loss of water? Authors are kindly asked to review and modify if appropriate.
Abbreviations should be reviewed.
Perhaps the main problem of the article is that the Discussion section, despite being interesting and including relevant references, fail to synthesise in a clear manner the practical implications of the observations of the study. The way sodium transporters changes may have a significant impact should be discussed. I must confess that some of the data generated of the study were surprising. These ideas should be included in a Conclusion section. I know that this section is not mandatory but it will improve the clarity of the message for the readers. If not, these comments should at least be included in the Discussion section.
Round 2
Reviewer 2 Report
Comments and Suggestions for Authors
Thank you for making all the required modifications and clarifications in the article. I think the article will be very interesting for the readers. In this regard, I believe the article is worth to be published in the actual form.